# OpenReview forum: "SafeSearch: Automated Red-Teaming of LLM-Based Search Agents"
_ICML.cc/2026/Conference — ICML 2026 regular_

### Official Review · Reviewer_ctzm · 2026-03-12

**Soundness:** 3
**Presentation:** 3
**Significance:** 4
**Originality:** 3
**Overall Recommendation:** 4
**Confidence:** 3

**Summary:**

A central theme discussed by this paper is that search agents inherit a distinctive safety problem from the open web: even benign queries can produce unsafe answers when unreliable search results are surfaced and then trusted by the model. The submission proceeds to investigate a central area in search-agent safety by introducing SAFESEARCH, an automated red-teaming framework that generates 300 test cases across five risk types, evaluates three search-agent scaffolds over 17 LLMs, and shows that vulnerability remains high in many settings, while stronger orchestration such as deep-research-style scaffolds can reduce but not eliminate risk.

**Compliance With Llm Reviewing Policy:**

Affirmed.

**Key Questions For Authors:**

How sensitive are the headline conclusions to the choice of search service, especially less curated or less dominant providers than Google Search?

Can you provide a more direct calibration of how SAFESEARCH ASR correlates with failure rates observed in naturally occurring, non-injected search traces?

Since the evaluation pipeline uses LLMs for generation and judging, did you test cross-judge robustness with a materially different evaluator family?

How much of the scaffold safety gain comes from extra search budget versus better reasoning/orchestration per se?

**Limitations:**

Yes, they did.

**Strengths And Weaknesses:**

### Soundness

The paper is technically stronger than many benchmark-style submissions because it articulates a clear threat model, includes an explicit generate-filter pipeline for test construction, validates the checklist-based evaluator against human annotations, and performs a nontrivial set of ablations on injection position, timestamping, token limits, and generator choice. The main empirical result—that many search agents are highly susceptible to injected unreliable results, with scaffold choice materially affecting ASR—is supported by broad experiments. That said, the study still depends heavily on simulation assumptions and on LLMs inside nearly every part of the evaluation stack, so the strength of the claims should be interpreted as strong evidence about a realistic stress-testing regime rather than a definitive estimate of real-world deployed risk.

**Strengths**
- Clear and conservative threat model: benign queries, at most one unreliable website, agent-agnostic test cases.
- Quality filtering via attainability and integrity reduces the chance that “safe” behavior is really just infeasible test construction.
- Evaluator validation is reasonably convincing: reported human/LLM agreement is high, and ASR on benign settings is near zero on average.
- Broad benchmarking across models and scaffolds gives the results more weight than a narrow case study.

**Weaknesses**
- External validity remains the biggest concern: the setup injects one synthesized unreliable website into cached search results, which is plausible but still a stylized simulation.
- The framework relies on LLMs for test generation, website generation, safety evaluation, and helpfulness evaluation, which compounds model-dependent bias.
- Helpfulness is explicitly judged as perceived usefulness rather than factual correctness, which is reasonable for one lens but weakens interpretability of the safety/helpfulness tradeoff.
- Real proprietary systems are excluded, so the strongest deployment-facing claim is indirect.

### Presentation

The paper is well organized and generally easy to follow. The motivation is concrete, the threat model is spelled out clearly, the pipeline is intuitive, and the figures do real explanatory work rather than just decorating the narrative. The appendices are unusually useful: they clarify the RAG-vs-search distinction, provide prompt details, validate several design choices, and expose limitations. My main presentation concern is that some of the most decision-relevant caveats live in the appendix rather than the core text, and the paper occasionally blurs benchmark utility with real-world prevalence.

**Strengths**
- Strong structure from motivation to threat model to generation pipeline to experiments.
- Good qualitative figures and response examples help make failure modes legible.
- Related-work positioning is thoughtful, especially the distinction from RAG poisoning and prior search-safety work.
- Reproducibility appears relatively good, with prompts, configurations, and code/data release claims

**Weaknesses**
- Key caveats about simulation realism, search-service coverage, benchmark scale, and efficiency tradeoffs are deferred to later appendix discussion.
- “Helpfulness” could be more explicitly distinguished from correctness in the main narrative to avoid reader over-interpretation.
- The framing sometimes reads slightly stronger than the evidence warrants on deployed-system risk.

### Significance

This paper addresses an important and under-evaluated problem. Search-enabled agents are increasingly central to practical LLM deployment, and the shift from curated corpora to noisy, vendor-controlled web search materially changes the threat model. A framework that can systematically audit susceptibility to unreliable results, compare scaffolds, and probe defenses is likely to be useful to both researchers and practitioners. The main limitation is that the significance of the benchmark as a tool is clearer than the significance of the exact measured ASR values as estimates of real-world incidence.

**Strengths**
- Timely problem with immediate practical relevance.
- Useful empirical takeaway that scaffold design matters substantially, not just base-model choice.
- The framework appears valuable for iterative auditing, not only one-off benchmarking.
- The knowledge-action-gap finding is conceptually useful for future mitigation work.

**Weaknesses**
- Benchmark size is still modest at 300 curated cases.
- The paper stops short of proposing a strong new defense; filtering helps but remains incomplete.
- Significance for closed commercial systems is somewhat indirect because they are not directly evaluated.

### Originality

The originality is good, though not radical. The paper is not the first to study indirect prompt injection, misinformation, or external-context unreliability, and it overlaps with recent work on unsafe search-augmented systems. Its main novelty is in the combination and framing: agent-centric rather than system-outcome-centric evaluation, benign-query red-teaming rather than risk-inducing-query construction, unified coverage of multiple risk types, and a reusable simulation-based pipeline that explicitly targets the search-agent setting rather than standard RAG. That is enough to make the contribution feel meaningfully new, even if it is more of a benchmarking/methodology advance than a foundational algorithmic one.

**Strengths**
- Nicely distinguishes search-agent threats from RAG threats.
- Unified benchmark over five heterogeneous risk categories is a useful extension over narrower prior work.
- Benign-query framing is a valuable shift from more obviously adversarial setups.

**Weaknesses**
- The core technical ingredients are largely compositions of existing ideas: simulation-based testing, LLM-as-judge, synthetic data generation, and prompt-based filtering.
- The paper’s novelty is strongest at the evaluation-framework level, not at the level of new safety mechanisms.

---

> ### Author Rebuttal · Authors · 2026-03-30
>
> We thank the reviewer for the helpful comments!
>
> **C1. Concerns in weaknesses: gap from deployed-risk estimate, implications for commercial systems, and validity of the injection setup.**
>
> **RC1**: We try our best to address key points within the space limit; if we missed anything important, please let us know in the later round.
> - **Scope**: SafeSearch exposes agent-centric vulnerabilities under a plausible threat model and provides actionable safety signals before deployment. Such red-teaming is essential for both open-source and commercial systems, as also noted in OpenAI's Deep Research system card.
> - **Realism**: Though simulation-based, SafeSearch is designed to reflect deployed risk closely. Appendix J.3 discusses how it facilitates addressing real-world threats. The injection setup is supported by **seven experiments in Appendix F** and **two new analyses** (website generator and judge model; see **RC2 to Reviewer TDEB**). **RC3** below further shows that SafeSearch ASR reflects natural safety failures. Together, these support the setup’s representativeness.
>
> **C2. "How sensitive are the headline conclusions to the choice of search service, especially less curated or less dominant providers than Google Search?"**
>
> **RC2**: We newly tested two less dominant services, Jina and Tavily. ASR varies only modestly, so the main conclusion is not Google-specific.
>
> **Table 1. ASR (%).** Columns: tool-call agent backbends.
> ||GPT-4.1-mini|Qwen3-235B|
> |-|-|-|
> |Google|66.56|29.22|
> |Jina|69.11|26.56|
> |Tavily|66.89|27.22|
>
> **C3. "Can you provide a more direct calibration of how SAFESEARCH ASR correlates with failure rates observed in naturally occurring, non-injected search traces?"**
>
> **RC3**: Thank you for the thoughtful comment.
> - Direct calibration is hard because, unlike injected traces, non-injected traces lack a ground-truth checklist of unsafe behaviors. Meanwhile, we do understand your concern about how SafeSearch ASR reflects real-world safety failures. As a best-effort study, we conduct a sanity check.
> - **Experiment**. Using the same 300 queries without injection, a tool-call agent (Qwen3-235B) generates 900 responses, evaluated with the injected case's safety checklist as a **proxy**. 8/900 responses (0.88%) are flagged. Manual review shows 7/8 over-endorse products (advertisements), and 1/8 hallucinates after a failed search (no related results). This suggests the evaluator is not arbitrarily over-flagging and that SafeSearch ASR **meaningfully reflects real-world safety failures**.
>
> **C4. "Since the evaluation pipeline uses LLMs for generation and judging, did you test cross-judge robustness with a materially different evaluator family?"**
>
> **RC4**: We conducted a cross-judge robustness analysis by sampling 500 agent responses from **RC2** and evaluating them with **five safety judge models**. ASRs are similar across them, suggesting only modest judge dependence, likely because the safety evaluation is checklist-based.
>
> **Table 2. ASRs by judge models.**
> |Evaluator|ASR(%)|
> |-|-|
> |GPT-4.1-mini|46.8|
> |DeepSeek-V3.1|47.4|
> |Gemini-2.5-Pro|46.8|
> |Gemini-3-Flash|45.0|
> |Qwen3-32B|46.4|
> |5-JudgeMajority|47.6|
>
> **C5. "How much of the scaffold safety gain comes from extra search budget versus better reasoning/orchestration per se?"**
>
> **RC5**: We ran new experiments.
> - **Answer**. Both factors matter. For the best-performing deep research in the paper, its main advantage appears to **come from the ability to use the available budget more effectively**.
> - **Setup**. To isolate budget effects, we implemented budget-forced variants: *tool-call forced* and *deep research forced*. Both variants consume exactly one search per turn and fully exhaust the search budget across turns. We tested budgets of 3/6/9, with a 1,000-token per-webpage cap to stay within max context limits.
>
>     **Table 3: ASR (%) (backend: GPT-4.1-mini).** Columns: budget.
>     ||3|6|9|
>     |-|-|-|-|
>     |tool-call agent (old)**->4th**|74.56|74.00|74.33|
>     |tool-call forced **->1st**|49.89|43.67|**36.67**|
>     |deep research (old)**->3rd**|63.22| 58.29|57.00|
>     |deep research forced **->2nd**|59.44|56.44|46.16|
>
> - **Budget-forced vs. autonomous**. At matched budgets, *tool-call forced* is consistently safer than *deep research forced*, showing its superiority. Under autonomous settings, *deep research (old)* **conversely** outperforms *tool-call agent (old)* at all budgets.
> - **Why so?** Generally, more searches lead to better safety, consistent with findings in Section 4.3. A closer look reveals the **budget under-use of autonomous agents**. The tool-call agent can be a simple yet safer response finalizer, but it lazily uses the available search budget (typically only 1 call). Deep research is better in this aspect. Once search advantage is controlled, deep research no longer outperforms the tool-call agent.
>
> If accepted, we will incorporate these results and refine the statements in the main text, given the expanded camera-ready page limit.

---

> > ### Author Rebuttal · Reviewer_ctzm · 2026-04-04
> >
> > The authors have addressed my concerns; I maintain my score of recommending weak acceptance.

---

> > > ### Author Response · Authors · 2026-04-04
> > >
> > > Dear Reviewer ctzm,
> > >
> > > Thank you very much for your careful review, your encouraging initial assessment, and your post-rebuttal acknowledgement that our responses fully addressed your concerns. We especially appreciate your recognition of the paper's strengths, including the clear threat model, broad benchmarking, and practical relevance, and we are glad the rebuttal helped clarify the main issues you raised.
> > >
> > > **Summary of rebuttal**. In response to your questions, we carried out several additional analyses: experiments on two additional search backends (Jina and Tavily), a sanity-check study on non-injected search traces, a cross-judge robustness evaluation across five evaluator families plus majority vote, and budget-controlled comparisons between tool-call and deep-research scaffolds. We believe these new results further strengthen the paper’s claims on backend robustness, real-world relevance, judge dependence, and the role of search budget versus orchestration, and we will incorporate them in the final version.
> > >
> > > Thank you again for the consistently positive and constructive feedback throughout the process. Given that you noted your concerns were fully resolved, we would be very grateful if you would consider whether the additional evidence merits a score increase or confidence adjustment.
> > >
> > > Sincerely

---

### Official Review · Reviewer_TDEB · 2026-03-13

**Soundness:** 3
**Presentation:** 3
**Significance:** 4
**Originality:** 3
**Overall Recommendation:** 5
**Confidence:** 3

**Summary:**

The paper introduced SafeSearch, an automated red-teaming framework to evaluate the safety of LLM-based search agents when exposed to unreliable websites. The framework maintains benign user queries and injects a single unreliable webpage into the search results, the measure if the agent’s final answer shows one of 5 target behaviors. The framework includes automated test generation, a filtering step to ensure attainability and integrity, and automated evaluation. The authors build a benchmark consisting of 300 queries, used to test 3 scaffolds across 17 LLMs.

The authors find that agents can be highly vulnerable to unreliable context, with the highest ASR being 90.5%. The paper also shows high variation depending on the underlying LLM and the scaffold, with stronger reasoning models being more robust.

**Compliance With Llm Reviewing Policy:**

Affirmed.

**Key Questions For Authors:**

1. How much of the scaffold advantage remains after controlling for total search budget and effective context length?
2. How sensitive are the conclusions to the exact injection policy, especially ranking position, snippet style, and whether the unreliable page appears in later search rounds as well?

**Limitations:**

yes

**Strengths And Weaknesses:**

## Strengths

- The authors tackle an important and timely problem given the proliferation of web-search agents. The framing is convincing and important.
- The threat model is clear and the automated generation, quality filtering and evaluation protocol are scalable.
- I also find the perspective of keeping the user query benign, but modifying the context (websites) interesting and realistic.
- The evaluation is thorough, covering 3 scaffolds and 17 LLMs. The authors also present several ablations.
- The defense analysis is useful because it shows that prompt-only mitigations are weak while filtering helps only partially.

## Weaknesses

- The evaluation depends on a simulated injection procedure in which one unreliable website is appended to authentic results, usually on the first search round. This is a reasonable and sensible approximation, but it still only partially captures the messiness of the real world.
- While the reliance on LLMs in multiple parts of the framework makes it more scalable, this can also be an issue. The human agreement result helps, but some risk of correlated bias remains because the same general family of tools is involved throughout the workflow.

---

> ### Author Rebuttal · Authors · 2026-03-30
>
> We thank the reviewer for the helpful comments!
>
> **C1. "The evaluation depends on a simulated injection procedure in which one unreliable website is appended to authentic results, usually on the first search round. This is a reasonable and sensible approximation, but it still only partially captures the messiness of the real world."**
>
> **RC1**: We do understand your concern about red-teaming representativeness. In Appendix F, we show that injecting the unreliable website earlier or in the first search position yields higher ASRs than our default setup, suggesting that the red-teaming findings are not specific to this concrete setup. In real-world cases, the threat could be even messier, but **even under this conservative setup, most search agents remain highly vulnerable**. Beyond these, Appendix J.3 further discusses our efforts toward simulation realism.
>
> **C2. "While the reliance on LLMs in multiple parts of the framework makes it more scalable, this can also be an issue. The human agreement result helps, but some risk of correlated bias remains because the same general family of tools is involved throughout the workflow."**
>
> **RC2**: The paper already includes two checks in Appendix F, suggesting that our conclusions do not depend on one LLM family:
> - **Test case generator**: Replacing o4-mini with Qwen3-8B still produces effective test cases; ASRs drop only moderately rather than collapsing.
> - **Baseline agent for filtering**: Replacing Qwen3-8B with GPT-5-mini retains more challenging test cases.
>
> In the rebuttal, we have two new experiments.
> - **Website generator**: Replacing GPT-4.1-mini with Gemini-2.5-Flash yields similar ASRs, suggesting limited sensitivity to this component.
>
>     **Table 1. ASR (%)**. Rows: website generators; Columns: backend models of the tool-calling agent.
>     | |GPT-4.1-mini|Qwen3-235B|
>     |-|-|-|
>     |GPT-4.1-mini|66.56|29.22|
>     |Gemini-2.5-Flash|70.97|30.59|
>
> - **Safety evaluator** (kindly see **RC3** to reviewer ctzm for details): Evaluating 500 responses with five judge models, including Gemini-2.5-Pro and GPT-4.1-mini, shows minimal dependence on judge choice.
>
> We will make these robustness checks more explicit in the revision.
>
> **C3. "How much of the scaffold advantage remains after controlling for total search budget and effective context length?"**
>
> **RC3**: We ran new experiments.
> - **The answer**. Under controlled search budgets, **the scaffold advantage of deep research is reversed** and the tool-call agent conveys safer responses. To clarify, the main advantage of deep research is largely its ability to call more searches.
> - **Setup**. To isolate budget effects, we implemented budget-forced variants: *tool-call forced* and *deep research forced*. Both variants consume exactly one search per turn and fully exhaust the search budget across turns. We tested 3 search budget scales (3/6/9) with a 1,000-token limit per webpage to avoid exceeding the max context size. The results are as follows.
>
>     **Table 2: ASR (%) (backend: GPT-4.1-mini).** Columns: budget.
>     ||3|6|9|
>     |-|-|-|-|
>     |tool-call agent (old)**->4th**|74.56|74.00|74.33|
>     |tool-call forced **->1st**|49.89|43.67|**36.67**|
>     |deep research (old)**->3rd**|63.22| 58.29|57.00|
>     |deep research forced **->2nd**|59.44|56.44|46.16|
>
> - **Budget-forced vs. autonomous**. At matched budgets, *tool-call forced* is consistently safer than *deep research forced*, showing its superiority. Under autonomous settings, *deep research (old)* **conversely** outperforms *tool-call agent (old)* across budgets.
> - **Why so?** Generally, more searches lead to better safety, consistent with findings in Section 4.3. A closer look reveals the **budget under-use of autonomous agents**. The tool-call agent can be a simple yet safer response finalizer, but it lazily uses the available search budget (typically only 1 call). Deep research is better in this aspect. Once search advantage is controlled, deep research no longer outperforms the tool-call agent.
>
> If accepted, we will incorporate these results and refine the corresponding statements in the main text, given the expanded camera-ready page limit.
>
> **C4. "How sensitive are the conclusions to the exact injection policy, especially ranking position, snippet style, and whether the unreliable page appears in later search rounds as well?"**
>
> **RC4**: We had related studies in Appendix F.
> - **Ranking position**: Injecting the website first (94.6%), random-middle (90.1%), or last (90.5%) **does not substantially change ASR**.
> - **Snippet style**: Varying the **maximal token length** of the injected website and **enforcing persuasiveness** in website generation have further potential to improve attack success quantitatively. Our setting in this direction is conservative.
> - **Later search rounds**: We show that success is **not specific to one injection round**, and that our default round-1 injection is conservative, since later-round injection can yield even higher ASR.

---

> > ### Author Rebuttal · Reviewer_TDEB · 2026-04-03
> >
> > The authors have addressed my questions. I would love to see an expanded explanation of RC3 in the final paper. I maintain my score.

---

> > > ### Author Response · Authors · 2026-04-04
> > >
> > > Dear Reviewer TDEB,
> > >
> > > Thank you very much for your thoughtful review and for the consistently positive assessment of our work. We especially appreciate your recognition of the paper’s importance, the realism of the threat model, the scalability of the framework, the thorough evaluation, and the value of the defense analysis. We are also very grateful that you found our rebuttal to have fully resolved your concerns.
> > >
> > > **Summary of rebuttal**. In response to your feedback, we conducted several additional experiments in the rebuttal, including robustness checks with different website generators and judge models, controlled-budget experiments clarifying that the apparent scaffold advantage is largely driven by search-budget usage, and further analyses of injection policy showing that our default setup is conservative. As you suggested, we will faithfully incorporate these new results and provide an expanded explanation of RC3 in the final version if the paper is accepted.
> > >
> > > Thank you again for your constructive feedback and support throughout the process. If you feel it is appropriate given that your concerns have been fully addressed, we would be very grateful if you would consider a score increase or confidence adjustment.
> > >
> > > Sincerely

---

### Official Review · Reviewer_P8Ba · 2026-03-13

**Soundness:** 2
**Presentation:** 1
**Significance:** 2
**Originality:** 2
**Overall Recommendation:** 2
**Confidence:** 4

**Summary:**

The paper researches LLMs' unsafe search behavior in a red-teaming setting. Their major contribution is a framework for safety evaluation of search agents.

**Compliance With Llm Reviewing Policy:**

Affirmed.

**Final Justification:**

I thank the authors' effort and time investment. I appreciate their active rebuttal participation. I hold my original judgement on this work. Given the limited time period, my concern remains. I welcome any open public debate, and I will try to participate for the rest of the review process.

Thanks all.

**Key Questions For Authors:**

Please refer to the last section.
## Minor Issues
I choose not to enumerate all minor issues, as this will largely be spotted by the authors or editors.
I want to emphasize some of the misusage of phrase such as "non-trivial frequency". The authors may need more proofreading by themselves. Also, please ensure all abbreviations are defined before used. For example, ASR.

**Limitations:**

The authors discussed limitations in Appendix.

**Strengths And Weaknesses:**

## Strengths
To me, the major contributions of this paper include the scale of experiments, specifically, the number of LLMs they have evaluated, plus the proposed framework.

## Weaknesses
First, I know that the authors proposed an automated framework for red-teaming. However, the claimed contributions such as "threat model and test-case structure", "automated generation of test cases", and "injection setup", are largely engineering integration. The scientific contribution is limited.
Therefore, I believe the work is more suitable as a submission to benchmarking / dataset track.
And I will list the problems I concern, given the configuration of the paper.
Methodology-wise:
- 1. What is the definition of "low quality, misleading, manipulable" content? I think a clear definition is critical to understand the scope and the experimental results of this work, if their contribution is on benchmark/the efficacy of the framework.
- 2. Please clearly state why the framework is cost-efficient (in terms of token-cost? / dollar bill? / time budget?). I would appreciate either theoretical statements or empirical results. So far I did not see any. Please enlighten me if I missed any critical, compelling evidence.
Then, I concern about the experiment setup as follows:
- 1. What is the necessity of having "LLM w/ search", "LLM w/ tool calling"? To me, they are essentially the same under the paper's setup. And essential information is not fully stated. For example, for w/ search, in the experiments, how often do "LLMs w/ search" need the search operation?
- 2. What is the necessity of curating the dataset for evaluation? Is there any well-recognized benchmark? Overall, I think it is hard for me to learn a clear picture of "to what extend the framework is superior" and "why this work is innovative to the community". Yet, I still appreciate the amount of their experiment.
- 3. I think the claim of "harmless" is too bold. Based on my understanding, the "harmful" injection is done in an offline manner, and it is obvious that it will never be harmful to the real world users. Conducting adversarial procedure in a somehow sandbox, I think this is a regular implementation routine.

---

> ### Author Rebuttal · Authors · 2026-03-30
>
> We thank the reviewer for the helpful comments!
>
> **C1. "However, the claimed contributions ... are largely engineering integration. The scientific contribution is limited. Therefore, I believe the work is more suitable as a submission to benchmarking/dataset track."**
>
> **RC1**: We apologize for any misunderstanding caused by not clearly enumerating the paper's contributions in the Introduction. We respectfully clarify that the paper did not intend to claim the individual components (e.g., `test-case structure`) as standalone contributions. Rather, the technical contribution lies in effectively orchestrating them into a testing framework for an important, understudied threat to search agents.
>
> **Our contributions**. We thank the reviewer for recognizing the paper's benchmarking value. However, the paper goes beyond a static dataset by revealing the current limitations of agentic models and directions for improvement. In this sense, it is **similar in spirit to the scientific contributions of excellent ICML works [1–3]** that also advance model evaluation or safety. In revision, we will state the main contributions as: (1) formulating a new safety problem for search agents; (2) proposing an automated framework for scalable red-teaming; and (3) conducting comprehensive experiments that provide insights into search-agent safety.
>
> **References**:
> - [1] ITBench: Evaluating AI Agents across Diverse Real-World IT Automation Tasks. ICML 2025 Oral
> - [2] SWE-Lancer: Can Frontier LLMs Earn $1 Million from Real-World Freelance Software Engineering? ICML 2025 Oral
> - [3] TRUST-VLM: Thorough Red-Teaming for Uncovering Safety Threats in Vision-Language Models. ICML 2025
>
> **C2. "What is the definition of "low quality, misleading, manipulable" content?..."**
>
> **RC2**: Here, "low quality," "misleading," and "manipulable" are informal descriptors of the broader phenomenon of unreliable search results, not formal categories. Since search result reliability is context-dependent and difficult to define exhaustively, we **scope the paper around five representative risks**: indirect prompt injection, harmful output, misinformation, bias-inducing content, and sponsored advertisements. We define these in Appendix Table 2 and provide examples in Appendix K, and we will point readers to them more clearly in the revision.
>
> **C3. "Please clearly state why the framework is cost-efficient..." & "I think the claim of "harmless" is too bold..."**
>
> **RC3**: We are deeply sorry for causing the confusion.
> - When describing SafeSearch (e.g., line 69), our use of `harmless` and `cost-efficient` is **relative to real-world poisoning of search services**, which would require manipulating live rankings through costly and risky SEO during testing. SafeSearch instead studies the same downstream threat offline by injecting one unreliable result, making evaluation cheap and not affecting ordinary users.
> - **Evidence for cost-efficiency**. For a concrete estimate, reproducing SafeSearch testing for a tool-call agent (gemini-2.5-flash) on 300 test cases costs **$8.53** total, excluding relatively small search costs. The cost is arguably acceptable for developers. Much of it ($6.41/8.53) is a one-time cost, since test cases and unreliable webpages can be reused across agents. The framework is also **time-efficient**: human labeling alone takes 6.5 hours for 200 responses on average (Appendix G), whereas the LLM judge supports much higher throughput.
> - We will replace "harmless" with "sandboxed" to clearly convey how SafeSearch achieves harmless testing.
>
> **C4. "What is the necessity of having "LLM w/ search", "LLM w/ tool calling"? ..."**
>
> **RC4**: The `LLM w/ search workflow` and `LLM w/ tool calling` settings are distinct (details in Section 4.1 and Appendix D.1):
> - In the `search workflow` setting, the agent **passively** receives results from **one single search of the user query** and answers based on them.
> - In the `tool calling` setting, the agent has **more autonomy**, deciding how to search across **multiple rounds**. The concrete search queries are decided by the model.
>
> As shown in Table 1, `GPT-4.1-mini` achieves 90.5% ASR in the `search workflow` but only 77.8% in `tool calling`, highlighting the importance of including the two distinct scaffolds.
>
> **C5. "What is the necessity of curating the dataset for evaluation? ..."**
>
> **RC5**: As detailed in Section 2.2 (Related Works), **no existing benchmark adequately addresses the threat studied in our work**. We therefore propose the framework and curate a dataset to fill this gap, enabling a systematic study of search-agent safety. Existing benchmarks like GAIA and SimpleQA focus on task performance, while agent-safety benchmarks typically address narrower threats around search agents.
>
> Our contribution and novelty are **not about generic superiority**, but about providing a necessary testing framework.
>
> ---
> Thank you for the feedback on presentation. We will improve these in the revision.

---

> > ### Author Rebuttal · Reviewer_P8Ba · 2026-04-03
> >
> > ## My response
> > My concerns still remain. I think the authors agree that the contribution is about the highly engineered framework. And individual components are largely out there, and their framework has an incremental strength building upon it.
> >
> > However the complexity of this incremental combination is not really technical. For example, it is hard for me to identify the unique contribution of this paper, given similar priors like Luo et al. (2025) and Ou et al. (2025).
> >
> > The work still has some places which are logically debatable, maybe i didn't make it clear enough in the previous reply, I reiterate them here:
> > - The pipeline seems circular: test cases are LLM-generate / injected, websites are LLM-generated and evaluation is LLM-based. And the authors mention: passing the baseline agent does not imply clean behavior for other agents. So it seems like the benchmark is not only measuring model, it is also shaped by the framework's own design choices.
> > - The paper reports high human-LLM agreement (95–98%) as validation of the safety evaluator. However, the human annotators were the paper's own authors (if my understanding is right), who also designed the evaluation checklists being used for annotation. This creates a circular validation. Given the nature of the task (safety) is very subjective, is this setting reasonable to this part of the experiment? (Given this is an over 40 pages paper, I may miss some information, please correct me if I was wrong.)
> > - Is there any particular reason to inject unreliable website only in the first search call for the multi-turn cases? If your analysis on this part requires the websites to be non-stealthy. I also don't think stealthiness matters that much in this paper. And this part is far away from the reality.
> > - The paper already mentioned that the efficiency and operational cost are not evaluated and left for the future, then i guess the cost-efficiency shouldn't be a part of the contribution.
> > - Is there really no red-teaming framework worthy a comparison? If so, what makes them unable to be adopted to this problem setup?
> >
> > My other concern also includes: there are too many variables / dynamics can influence the experimental results, which is really hard for me to scrutinize the solidity of experimental details. For example, injection position, number of search results, token for website cap, temperature, etc. Some variables seem weak, for example, integrity filter is based on the baseline agent, but the appendix said different LLM may change the behavior, it may indicate an effect on a specific agent, not for the general cases.
> >
> > I maintain and defend my review.
> >
> > Luo et al. (2025) https://www.usenix.org/conference/usenixsecurity25/presentation/luo-zeren
> >
> > Ou et al. (2025): https://arxiv.org/abs/2510.09689

---

> > > ### Author Response · Authors · 2026-04-04
> > >
> > > Thank you for the detailed feedback.
> > >
> > > **FC1. I think the authors agree that the contribution is about the highly engineered framework. ...**
> > >
> > > **RFC1.** Sorry for causing your misunderstandings. To clarify, we emphasize again that the technical contribution lies in the effective orchestration of a red-teaming framework that addresses challenges faced by traditional testing methods in this new threat, while also being systematic, scalable, and cost-efficient. For a more detailed explanation of the contribution, we refer you to **RC1**.
> > >
> > > **FC2. ... For example, it is hard for me to identify the unique contribution of this paper, given similar priors like Luo et al. (2025) and Ou et al. (2025).**
> > >
> > > **RFC2.** Beyond the effectiveness of the SafeSearch red-teaming, this work makes a significant contribution by focusing on the safety of LLM-based search agents in scenarios involving unreliable search websites—--an area that has received limited attention. Section 2.2 (Lines 134-144) offers a detailed comparison with the two cited works, discussing differences in threat models, agent-centric behaviors, and risk coverage.
> > >
> > > **FC3. The pipeline seems circular: test cases are LLM-generated / injected, websites are LLM-generated and evaluation is LLM-based. ..., it is also shaped by the framework's own design choices.**
> > >
> > > **RFC3.** We kindly refer you to the **RC2** to Reviewer TDEB for the ablation of the LLM choices, which demonstrates that our careful design choices ensure a representative testing environment for search agents. We also encourage you to review **RC4** to Reviewer TDEB for comprehensive experiments confirming the representativeness of our testing methodology.
> > >
> > > **FC4. The paper reports high human-LLM agreement (95–98%) as validation of the safety evaluator. ... Given the nature of the task (safety) is very subjective, is this setting reasonable to this part of the experiment?**
> > >
> > > **RFC4.** Sorry for causing your misunderstandings: The evaluation checklists are automatically generated by the test generator, not manually crafted. In fact, using a checklist-assisted evaluation is specifically designed to address potential subjectivity in judgment. We refer you to **RC4 to Reviewer ctzm** for a discussion on the calibration advantages of this approach.
> > >
> > > **FC5. Is there any particular reason to inject unreliable website only in the first search call for the multi-turn cases? ... And this part is far away from the reality.**
> > >
> > > **RFC5.** We address this in Section 3.3 (Line 247), where we explain that the agent is given an opportunity to down-weight or disregard the injected website in subsequent calls. This design choice ensures that we do not overestimate real-world threats. We have found that injecting unreliable websites in later rounds results in higher ASRs. This is explicitly guided in Section 4.1 (Experimental setup, Line 340), with supporting data provided in Experiment 5 in Appendix F (Line 1037).
> > >
> > > **FC6. The paper already mentioned that the efficiency and operational cost are not evaluated and left for the future, then i guess the cost-efficiency shouldn't be a part of the contribution.**
> > >
> > > **RFC6.** Sorry for causing misunderstandings. The future work referenced relates to the explicit evaluation metric for search agent efficiency, beyond helpfulness score and ASR. The cost-efficiency advantage, on the other hand, lies in the testing framework itself, e.g., how to achieve the scenario of unreliable search results. We have provided empirical evidence supporting this claim in **RC3**, as requested.
> > >
> > > **FC7. Is there really no red-teaming framework worthy a comparison? If so, what makes them unable to be adopted to this problem setup?**
> > >
> > > **RFC7.** As confirmed in **RC5** and **RFC2** above, SafeSearch addresses a fundamentally novel threat model: safe queries combined with unreliable search results. Existing works do not address this particular problem, and our proactive injection method strictly represents this.
> > >
> > > **FC8. there are too many variables / dynamics can influence the experimental results, which is really hard for me to scrutinize the solidity of experimental details. For example, injection position, number of search results, token for website cap, temperature, etc. Some variables seem weak, for example, integrity filter is based on the baseline agent, but the appendix said different LLM may change the behavior, it may indicate an effect on a specific agent, not for the general cases.**
> > >
> > > **RFC8.** To clarify, we have rigorously validated red-teaming setup with **seven experiments in Appendix F and two new analyses (website generator and judge model; see RC2 to Reviewer TDEB)**. Meanwhile, we clarify that the baseline agent is for filtering highly usable test cases, while the safety metric ASR is computed at the dataset level and represents a stronger empirical safety level, with exactly 900 runs for testing each search agent.
> > >
> > > We appreciate your valuable efforts and time for reviewing our work.

---

### Decision · Program_Chairs · 2026-04-30

**Decision:**

Accept (regular)

**Comment:**

The paper proposes an automated red-teaming framework to evaluate the safety of LLM-based search agents. The studied problem is important and challenging. The framework design is novel, where it can automatically modify the context (websites) while keeping the users' prompt intention, so that even a small sample of test cases can effectively evaluate LLM agents' capability. The paper surely can be more suitable for the benchmark track, but providing a testing framework can be an important contribution to the community.